# UAV Formation Control With External Obstruct Based On Differential Game*

1st Zhibin Yang
*the School of Science*
*Liaoning University of Technology*
*Jinzhou, China*
yhvibb0511@163.com

2nd Yang Chen
*the School of Science*
*Liaoning University of Technology*
*Jinzhou,China*
chenyangedu@163.com

3rd Lei Liu
*the School of Science*
*Liaoning University of Technology*
*Jinzhou,China*
liuleill@live.cn

*Abstract*—In this paper, the problem of multi-UAV formation control with external interference is studied. The indicators of each drone in the drone formation are independently selected based on its local information, and each drone tries to minimize its own performance indicators. Because the target of each UAV in the formation is different, the formation control problem in this paper is expressed as a differential game problem. In the actual flight of UAV formation, there will be some physical interference. This paper uses a commonly used UAV point mass model and designs an effective interference observer to estimate the negative effects of external interference. After eliminating the external interference, it is helpful to achieve the expected formation control goal better. Finally, a novel open-loop Nash strategy design method is used to estimate the terminal state of the UAV to achieve the complete distribution of the UAV.

*Index Terms*—multi-drone formation, differential game, interference observer

## I. INTRODUCTION

UAV has many functions, and is very flexible and lightweight, and has many applications in many fields. With the rapid development of UAV technology, it is playing an increasingly important role in many fields. However, the load-bearing capacity, information processing capacity and task capability of a single UAV are weak. But if multiple drones can form a formation, it can overcome a series of problems faced by a single drone. Many teams at home and abroad have conducted a series of studies on the problem of UAV formation. Multi-UAV formation control refers to the coordination and control of UAVs in a multi-aircraft system to form a predetermined formation structure. The research in this field involves communication between UAVs, cooperative motion planning and robustness in dynamic random environments. Multi-UAV formation is widely used, including military warfare, logistics distribution, surveying and mapping fields. However, the complex and changeable battlefield environment also brings great challenges to UAV formation control. In recent years, there have been a lot of excellent achievements in the field of multi-UAV formation control. In [1], using the graph theory method, an output feedback linearization technique with consistency protocol is adopted to maintain the specified position of multiple UAVs on time-varying geometry. By introducing fault detection logic to transform

Identify applicable funding agency here. If none, delete this.

the interconnection topology, the controller can realize the stability of formation flight. In [2], the control rate of a fixed-wing UAV is designed, the decision-making process of UAV formation control is clarified, and the free conversion of flight state is realized. In [3], a physically understandable guidance rate for multi-UAV formation flight is designed using a virtual structure approach. In [4], a multi-UAV formation control strategy based on virtual double integral dynamics is proposed.

In the execution of time-limited tasks, UAVs will encounter a variety of unexpected situations and practical problems, such as wind speed, rainfall, fog weather, which will directly affect the control of UAVs formation. These are inevitable situations, many existing results have explored a lot of methods to eliminate the adverse effects of interference, one of the key problems in the current research is to design an interference observer to estimate and compensate for such external interference.

In recent years, differential game theory and multi-UAV formation control have attracted much attention in the field of modern control systems. In [5], a game problem of formation control is studied, which allows players in formation to keep a certain formation while avoiding obstacles. In [6], non-cooperative game is studied, and formation control is realized through Nash equilibrium strategy to ensure the stability of formation control. In [7], the differential game method is adopted to solve the control problem of robot formation. In [8], based on differential game, feedback controller is synthesized by rolling optimization method, and mobile robot formation control is carried out on the basis of graph theory. On the basis of previous studies, this paper selects a general UAV point mass model, establishes a multi-UAV formation, and studies a distributed multi-unmanned formation control based on differential game method.

## II. PROBLEM FORMULATION

### A. UAV Model

Now there are many kinds of UAV models on the market, the common UAV models are quadrotor UAV model, fixed wing UAV model.There are a variety of drone models on the market today, so it is impossible to have a universal drone modelm, the common UAV models include four-rotor UAV model and fixed-wing UAV model. In this paper, we adopt

the UAV model used in literature [9], which is a fixed-wing UAV model widely used in many literatures:

$$\begin{aligned}
\dot{x}_i &= V_i \cos\gamma_i \cos\chi_i \\
\dot{y}_i &= V_i \cos\gamma_i \sin\chi_i \\
\dot{h}_i &= V_i \sin\gamma_i
\end{aligned} \tag{1}$$

where $x_i, y_i, h_i$ represents the positions of the x, y and z axes in the inertial coordinate system, respectively, $V_i$ is the speed of the drone, $\gamma_i, \chi_i$ stands for heading Angle and track Angle respectively.

Assuming that the thrust direction is consistent with the vector direction, the dynamics model of the UAV point mass model described in (1) can be described as:

$$\begin{aligned}
\dot{V}_i &= \frac{T_i - D_i}{m_i} - g\sin\gamma_i \\[4pt]
\dot{\gamma}_i &= \frac{L_i \cos\phi_i - m_i \cos\gamma_i}{m_i V_i} \\[4pt]
\dot{\chi}_i &= \frac{L_i \sin x\phi_i}{m_i V_i \cos\gamma_i}
\end{aligned} \tag{2}$$

To describe the position of drone i in the coordinate system, we define $p_i = [x_i, y_i, z_i]$ , and $v_i = [\dot{x}_i, \dot{y}_i, \dot{z}_i]^T$ . Define $u_{x_i}, u_{y_i}, u_{z_i}$ as the actual control input, where $\ddot{x} = u_{x_i}, \ddot{y} = u_{y_i}, \ddot{z} = u_{z_i}$. Take $p_i$ and $v_i$ defined above as intermediate variables, and $v_i = \dot{p}_i$ , while we can define a new control input according to (5): From the above intermediate variables and, we can redefine the dynamics model of the fixed wing UAV used in this paper and add the effect of perturbation to it:

$$\begin{aligned}
\dot{p}_i &= v_i \\
\dot{v}_i &= u_i + c
\end{aligned} \tag{3}$$

In this paper, the leader-following model composed of N UAVs is used. Therefore, the leader and the following two types of UAVs are to be introduced into the influence of disturbance respectively.

*B. Information graph*

We can use graph theory to represent the exchange of information between drones. We define $G = (V, \varepsilon)$ as the infographic between drones, as the node of drone i, and $v_i \in V$ as the $e_{ij} \in \varepsilon$ directional information transmission from drone i to drone j. We first need to define each drone to be fully reachable on the infographic, and ensure that each drone is connected on the infographic.

*C. Nash equilibrium*

In [10], a distributed MPC framework is discussed to achieve Nash equilibrium strategy in a formation. Each drone in a formation is equipped with MPC, only exchanges information with neighboring drones, and then determines its own behavior according to the exchanged information, so each drone is independent from each other. Then, the requirements of the leader UAV and the following UAV in the formation can be expressed mathematically through the performance indicators respectively.

The performance metrics in this article consist of four parts:The formation required for step $0 - (N-1)$ energy consumption in the processThe terminal state penalty function and the predetermined velocity penalty function of step $N$ . The Nash equilibrium strategy of UAV is as follows: (refer to [11])

$$J_i(u_1^*, u_2^*, ..., u_N^*) \leq J_i(u_1^*, u_2^*, ..., u_i, ..., u_N^*) \tag{4}$$

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
