# OpenReview forum: "UAV Formation Control With External Obstruct Based On Differential Game*"
_IEEE.org/ICIST/2024/Conference — IEEE ICIST 2024 Conference Submission_

### Official Review · Reviewer_boRQ · 2024-08-22
**UAV Formation Control With External Obstruct Based On Differential Game**

**Rating:** 2
**Confidence:** 5

**Review:**

In this paper, the problem of multi-UAV formation control with external interference is studied. However, this article lacks innovation. And the writing structure of the article is not clear.

---

### Official Review · Reviewer_99Ww · 2024-08-22
**reject**

**Rating:** 3
**Confidence:** 5

**Review:**

The paper presents a approach to multi-UAV formation control under external interference, utilizing a differential game framework and a novel open-loop Nash strategy. However, this study does not introduce novel insights, and the organization of the writing is unclear.

---

### Official Review · Reviewer_kEz9 · 2024-08-24
**Review Comments for Manuscript NO.180**

**Rating:** 3
**Confidence:** 4

**Review:**

The manuscript  lacks substantial content in some critical sections, including theoretical explanations, detailed methodologies, and simulation results. A manuscript is expected to present a complete and thorough exploration of the research topic, including problem formulation, methodology, results, and discussion.
Without these essential components, the manuscript fails to provide a comprehensive understanding of the research conducted and its contributions to the field. Therefore, the manuscript, in its current form, is incomplete and does not meet the standards required for publication.

---

### Decision · Program_Chairs · 2024-09-08

Accept (Oral)